# Biochar as a Soil Amendment for Restraining Greenhouse Gases Emission and Improving Soil Carbon Sink: Current Situation and Ways Forward

**Ahmed Mosa** [1,*] **, Mostafa M. Mansour** [1]**, Enas Soliman** [1]**, Ayman El-Ghamry** [1]**, Mohamed El Alfy** [2] **and Ahmed M. El Kenawy** [3]

1  Department of Soils, Faculty of Agriculture, Mansoura University, Mansoura 35516, Egypt
2  Department of Geology, Faculty of Science, Mansoura University, Mansoura 35516, Egypt
3  Department of Geography, Mansoura University, Mansoura 35516, Egypt
*  Correspondence: ahmedmosa@mans.edu.eg

**Abstract:** The global exponential rise in greenhouse gas (GHG) emissions over the last few decades has triggered an urgent need to contextualize low-cost and evergreen technologies for restraining GHG production and enhancing soil carbon sink. GHGs can be mitigated via incorporating biochar into soil matrix to sequestrate the mineralized carbon in a stable form upon organic matter decomposition in soil. However, the efficiency of using biochar to offset GHG emissions from soil and terrestrial ecosystems is still debatable. Moreover, in the literature, biochar shows high functionality in restraining GHG emissions in short-term laboratory studies, but it shows minimal or negative impacts in field-scale experiments, leading to conflicting results. This paper synthesizes information on the ability of biochar to mitigate carbon dioxide ($CO_2$), nitrous oxide ($N_2O$), and methane ($CH_4$) emissions from soil and organic biomass, with an emphasis on cropland soils. The feedstock type, pyrolysis temperature, and application rate factors showed significant effects on controlling the effectiveness of biochar in restraining GHG emissions. Our study demonstrates that biochar, taken as a whole, can be seen as a powerful and easy-to-use tool for halting the rising tide of greenhouse gas emissions. Nonetheless, future research should focus on (i) identifying other indirect factors related to soil physicochemical characters (such as soil pH/EH and $CaCO_3$ contents) that may control the functionality of biochar, (ii) fabricating aged biochars with low carbon and nitrogen footprints, and (iii) functionalizing biologically activated biochars to suppress $CO_2$, $CH_4$, and $N_2O$ emissions. Overall, our paradoxical findings highlight the urgent need to functionalize modern biochars with a high capacity to abate GHG emissions via locking up their release from soil into the carbonaceous lattice of biochar.

**Keywords:** biochar; croplands and rangelands; carbon sequestration; organic manures

## 1. Introduction

The exponential increase in greenhouse gas (GHG) emissions following anthropogenic activities in the last few decades has caused a pronounced rise in climatic changes with consequent environmental crises in global warming, drought, salinity, biodiversity, and diseases [1]. Human agricultural practices account for about 13.5% of global GHG emissions, including 80.4 petagrams of $CO_2$ per year (11 times the current rate of fossil fuel combustion), and about 63% of the world's non–$CO_2$ GHG emissions, including 84% of global $N_2O$ and 54% of global $CH_4$ emissions [2]. Other indirect emissions, such as those from machinery and transportation, are also produced by common agricultural practices. The greenhouse effect, which raises the earth's temperature, is caused when the produced GHGs trap infrared (IR) radiation that is emitted from the earth's surface [3]. It was reported that seven countries (China, USA, India, Australia, Brazil, Canada, and Chile) contribute to producing more than 50% of the world's total soil emissions, and about

49% of the agricultural–related emissions [4]. Since irrigated agriculture is the dominant cultivation system in arid and semi-arid regions, large amounts of GHGs are emitted due to different agricultural practices including manure application, rice cultivation, enteric fermentation, burning crop residues, manure storage in the open air, and using energy for operating irrigated pumps [5]. Egypt's GHG emissions (as an example for arid countries) increased rapidly to more than 133% between 1990 and 2012, with a total emission of around 318 million tonnes eq. $CO_2$ [6].

Although agriculture led to substantial increases in GHGs, some agricultural practices could have significant impacts on reducing these emissions. Agriculture can make considerable contributions to mitigating GHG emissions in the atmosphere by (i) increasing soil organic carbon sinks, (ii) reducing the carbon and nitrogen footprint of soil organic amendments, (iii) recycling crop residues into value-added products instead of burning, and (iv) reducing GHG emissions generated during organic matter decomposition [7–9]. Pyrolysis has received attention recently as an effective method of treating organic waste and agricultural byproducts. Thermal processing of agricultural crop residues lowers waste volume and transportation costs while producing value-added products. Biochar has become a focal point of multidisciplinary study over the last decade as a solution to various worldwide challenges due to its high functionality, non-sophisticated processing, and renewability potential. Biochar is a charcoal-like substance made from pyrolyzed biomass intended for utilization as a soil improver. It has been credited with multiple benefits, including its abilities to improve the fertility and water-holding capacity of soil, protect water quality, capture greenhouse gases, generate carbon neutral energy, and increase agricultural output, as well as its contributions to carbon sequestration and the removal of pathogens [10–13]. The appropriate pyrolysis technology for biochar production should be considered based on the targeted field of application. Slow pyrolysis is the most common technique for biochar production, with a pyrolysis temperature ranging between 300 and 700 °C at a long residence time (300 to 7200 s) and low heating rate (0.1 to 1 °C/s) [14]. Fast pyrolysis is operated at high pyrolysis temperature within the range of 500–1200 °C at a high heating rate (10–200 °C/s) [15]. Flash pyrolysis, however, features a higher pyrolysis temperature (>900 °C) and heating rate (>1000 °C/s) [16]. Vacuum pyrolysis is another technique, in which biomass is converted under sub-atmospheric pressure (pyrolysis temperature, heating rate, and pressure are within the ranges of 300–700 °C, 0.1–1 °C/s, and 0.01–0.20 MPa, respectively) [17]. Hydro-pyrolysis is another technology, which involves using a high-pressure hydrogen atmospheric condition inside the pyrolysis reactor (pyrolysis temperature = 350–600 °C, heating rate = 10–300 °C/s, pressure = 10–17 MPa, and residence time > 60 s) [18]. Unlike other pyrolysis techniques, the heating energy in microwave pyrolysis penetrates the carbonaceous biomass and causes a vibration in their internal molecules [19].

The high functionality of biochar, including its physical properties (porosity, large surface area, and high water-holding capacity) as well as its chemical properties (abundance of oxygen-containing functional groups, surface charge, and pH-modulating effect), suggests its potential utilization in reducing GHG emissions [20]. However, a great deal of uncertainty remains surrounding the competitiveness of biochar with traditional soil amendments (e.g., compost and farmyard manures) given its low nutrient content and high pH value [21]. Additionally, the efficacy of pristine biochar for restraining GHG emissions under field-scale applications is still questionable. The wide range of variation between biochars depends upon the multiplicity of factors that underlie thermochemical conversion of organic biomass [22–24]. The novelty of this review relies on assessing the key factors that might impede the functionality of biochar in restraining GHG emissions. In addition, this review highlights the urgent need to develop fit-for-purpose forms of functionalized biochars tailored to improving soil carbon sink and restraining GHG emissions from soil matrix.

Several key factors control the properties of biochar and its effectiveness in reducing GHG emissions, including (i) feedstock type (e.g., agricultural wastes, sludge and manures,

algal biomass, and crustacean shell wastes), (ii) pyrolysis type (slow/fast pyrolysis, flash pyrolysis, microwave pyrolysis, vacuum pyrolysis, and hydro-pyrolysis), (iii) thermal processing protocol (e.g., heating rate, pressure and carries gas, residence time, and reactor design), and (iv) soil application rates and methods (e.g., broadcasting, in-furrow, or mixture with soil amendments) [25]. Consequently, biochar functionalization has emerged as a new trend, providing a roadmap for enhancing the competitiveness of biochar and its sustainable soil application for reducing GHG emissions.

In this review, we aimed to (i) summarize the recent research undertaken to remove $CO_2$, $N_2O$, and $CH_4$ emissions from soil and terrestrial ecosystems; (ii) review the anomalies and similarities among several investigations of factors affecting the efficiency of biochar in restraining GHG emissions; (iii) examine research done to lessen greenhouse gas emissions from the composting process; and (iv) highlight recent attempts undertaken to functionalize modern biochars with high capacity to achieve neutrality of carbon/nitrogen and net zero emissions. This review will help the academic/research community, as well as decision-making entities and environmental agencies, in establishing a decision-making framework for the large-scale application of biochar to mitigate GHG emissions. Overall, this review synthesizes data from different dimensions of biochar utilization for achieving carbon neutrality and reducing GHG emissions from agricultural ecosystems.

## 2. Bibliographic Data Collection

Books, book chapters, research articles, review articles, and proceedings were all scoured for this review. All selected sources were written in English and published within the last decade (2010 onwards). All of these articles came straight from reputable sources (e.g., Scopus, Web of Science, ProQuest, EBSCO, and JSTOR). Figure 1 shows a sample of the growing body of literature in the Scopus database on the topic of using biochar to mitigate agricultural GHG emissions. In our review, the International Biochar Initiative (IBI), the United States Department of Agriculture (USDA), and the European Environment Agency all contributed to the credibility of our review by providing official reports, statistics, and proceedings. Use of Get-Data Graph Digitizer (ver. 2.22, Russian Federation) allowed us to convert the data visualizations into corresponding numerical values. In this review, biochar + carbon dioxide emissions, biochar + nitrogen oxide emissions, biochar + chlorofluorocarbon emissions, biochar + soil + carbon dioxide emissions, biochar + soil + nitrogen oxide emissions, and biochar + soil + chlorofluorocarbon emissions were the initial search keywords. In addition, a number of meta-analyses were reported from a variety of published articles to establish a solid assessment of the current state of the potential use of biochar for limiting GHG emissions and the prospects for this direction.

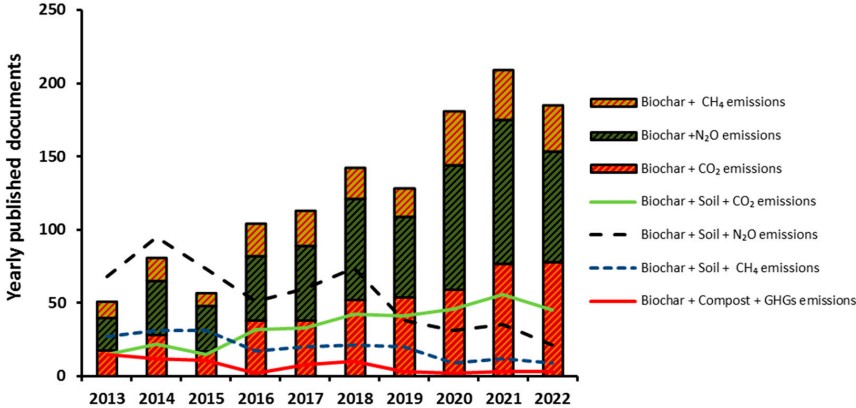

**Figure 1.** Number of documents published yearly in the Scopus database based on a query that employed the following keywords: biochar + $CO_2$ emissions, biochar + $N_2O$ emissions, biochar + $CH_4$ emissions, biochar + soil + $CO_2$ emissions, biochar + soil + $N_2O$ emissions, and biochar + soil + $CH_4$ emissions.

## 3. Effect of Biochar Application on Reducing GHG Emissions from Soil and Terrestrial Ecosystems

The research on biochar has attained significant momentum over the last decade. However, research related to field-scale applications has rarely been approached, highlighting the need for systematic efforts to fabricate end-use-appropriate biochars. The efficiency of biochar in reducing GHG emissions as compared with other soil amendments is summarized in Table 1. Endowed with unique functionality, biochar has been introduced as a promising adsorbent able to abate GHG emissions. Attempts have been undertaken to deploy biochar and its derivatives for mitigating $CO_2$, $N_2O$, and $CH_4$ emissions (Table 2). A comprehensive discussion of the mechanisms involved in $CO_2$, $N_2O$, and $CH_4$ emissions by biochar is provided below based on recent published results.

### 3.1. Effect of Biochar Application on Reducing $CO_2$ Emissions

Biochar exhibits a high potential for soil carbon sequestration given its high and resistant carbon content (particularly when its oxygen:carbon ratio is less than 0.2), prevalence of aromatic structures, and abundance of active functional groups, which improves its recalcitrant nature against decomposition in soil [33]. The carbon footprint of biochar ranges between 0.04 t$CO_2$eq (net emissions) and 1.67 t$CO_2$eq per t of biomass (net reduction) based on the feedstock type, system boundaries, and modalities of life cycle assessment studies [34]. Upon its soil application, biochar tends to change soil physical characteristics (e.g., water-holding capacity and bulk density) given its low skeletal density and higher surface area compared with particles of soil matrix [35,36]. The key role of biochar in modulating the pore structure parameters of soil (porosity, pore size distribution, connectivity, anisotropy, and fractal dimension) was a significant mechanism involved in restraining $CO_2$ emissions [37]. According to literature data, the overall average ability of biochar to reduce $CO_2$ emissions is −0.43%. This poor efficiency highlights the critical need to functionalize contemporary biochars with greater sorption capacity (Figure 2).

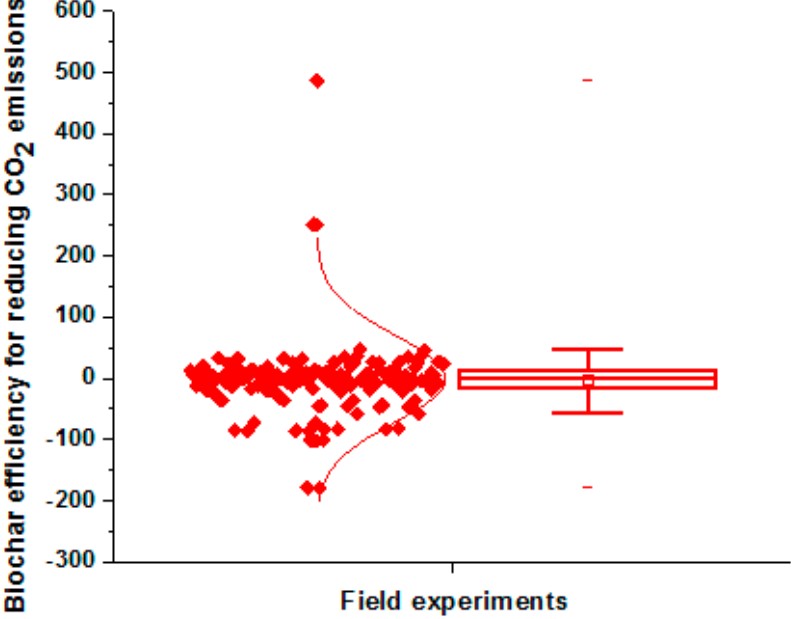

**Figure 2.** Biochar efficiency in reducing $CO_2$ emissions (%). Data are extracted from 38 field investigations comprising 165 individual observations. Box chart is illustrated by the mean (dot), median (centerline), lower and upper quartiles (the lower and upper borders of the box, respectively), and whiskers-error bars (the minimum and maximum observations).

**Table 1.** A comparision between biochar and other soil amendments in reducing GHG emissions.

| Country | Climatic Conditions | Soil Properties | | | Soil Amendments | Application Rate (Mg ha$^{-1}$) | Cultivated Crop | Years of Study | Yield (%) Compared with Control | GHGs Emission Rate Compared with Control (%) | | | Reference |
|---|---|---|---|---|---|---|---|---|---|---|---|---|---|
| | | Texture | pH | OC (g kg$^{-1}$) | | | | | | CO$_2$ | N$_2$O | CH$_4$ | |
| South Korea | - | Clay loam | 5.8 | 22.98 | Fly ash | 2.0 | Rice | 1 | 7.33 | - | 3.53 | −3.68 | |
| | | | | | Silicate slag | | | | 21.75 | - | 5.74 | 33.74 | |
| | | | | | Phosphogypsum | | | | 17.02 | - | 12.58 | 31.90 | |
| | | | | | Revolving furnace slag | | | | 20.57 | - | 7.73 | 26.99 | |
| | | | | | Blast furnace slag | | | | 13.00 | - | 8.83 | 9.82 | |
| Japan | - | Sandy loam | 6.1 | 21.32 | Biochar | 2.0 | Rice | 1 | 10.47 | - | 31.83 | −13.99 | [26] |
| | | | | | Silicate slag | | | | 25.58 | - | 17.65 | 14.68 | |
| | | | | | Phosphogypsum | | | | 23.26 | - | 14.88 | 20.14 | |
| | | | | | Biochar + azolla-cyanobacteria | | | | 27.91 | - | 26.30 | 7.85 | |
| | | | | | Silicate slag + azolla-cyanobacteria | 2.0 + 5.0 | | | 37.21 | - | 15.22 | 22.39 | |
| | | | | | Phosphogypsum + azolla-cyanobacteria | | | | 30.23 | - | 11.07 | 25.60 | |
| Bangladesh | - | Clay loam | 5.9 | 10.35 | Biochar | 2.0 | Rice | 1 | 15.85 | - | 20.00 | −9.49 | |
| | | | | | Silicate slag | | | | 28.05 | - | 14.18 | 18.35 | |
| | | | | | Phosphogypsum | | | | 24.39 | - | 9.87 | 27.22 | |
| | | | | | Biochar + azolla-cyanobacteria | | | | 25.61 | - | 25.06 | 9.49 | |
| | | | | | Silicate slag + azolla-cyanobacteria | 2.0 + 5.0 | | | 40.24 | - | 12.15 | 26.58 | |
| | | | | | Phosphogypsum + azolla-cyanobacteria | | | | 34.15 | - | 7.34 | 29.11 | |
| China | Warm temperate and semi-humid monsoon | Clay | 7.4 | 5.79 | Wheat straw | 7.5 | Soybean | 1 | 32.79 | - | −37.37 | - | [27] |
| | | | | | Pig manure | 15 | | | 24.59 | - | −49.49 | - | |
| | | | | | Cattle manure | 30 | | | 52.46 | - | −61.61 | - | |
| | | | | | Wheat straw | 7.5 | | 2 | 11.57 | - | −20.55 | - | |
| | | | | | Pig manure | 15 | | | 34.77 | - | −31.51 | - | |
| | | | | | Cattle manure | 30 | | | 64.81 | - | −31.51 | - | |
| | | | | | Wheat straw | 7.5 | Wheat | 1 | 7.25 | - | −5.74 | - | |
| | | | | | Pig manure | 15 | | | 3.63 | - | −30.33 | - | |
| | | | | | Cattle manure | 30 | | | 8.69 | - | −62.30 | - | |
| | | | | | Wheat straw | 7.5 | | 2 | 10.32 | - | −44.21 | - | |
| | | | | | Pig manure | 15 | | | 12.20 | - | −52.63 | - | |
| | | | | | Cattle manure | 30 | | | 16.87 | - | −67.37 | - | |
| Southeastern China | - | Silt loam | 6.5 | 18.10 | Steel slag | 8.0 | Early rice | 1 | 0.86 | 4.08 | 27.56 | 10.45 | [28] |
| | | | | | Biochar rice straw | 8.0 | | | 6.05 | 9.87 | 15.75 | −7.37 | |
| | | | | | Steel slag + biochar | 8.0 + 8.0 | | | 9.29 | 18.71 | 20.47 | 34.10 | |
| | | | | | Steel slag | 8.0 | Late rice | 1 | 1.49 | 12.25 | 27.40 | 14.89 | |
| | | | | | Biochar rice straw | 8.0 | | | 3.57 | 25.36 | 15.07 | 43.60 | |
| | | | | | Steel slag + biochar | 8.0 + 8.0 | | | 6.98 | 21.32 | 20.55 | 33.43 | |

Table 1. *Cont.*

| Country | Climatic Conditions | Soil Properties | | | Soil Amendments | Application Rate (Mg ha$^{-1}$) | Cultivated Crop | Years of Study | Yield (%) Compared with Control | GHGs Emission Rate Compared with Control (%) | | | Reference |
|---|---|---|---|---|---|---|---|---|---|---|---|---|---|
| | | Texture | pH | OC (g kg$^{-1}$) | | | | | | CO$_2$ | N$_2$O | CH$_4$ | |
| India | Sub-tropical, semi-arid | Loam | 8.1 | 5.90 | Azolla<br>Blue-green algae (BGA)<br>Azolla + BGA<br>Hyphomicrobium facile (A)<br>Burkholderia sp. (B)<br>Methylobacteruim oryzae (C)<br>A + B + C | -<br>-<br>-<br>-<br>-<br>-<br>- | Rice | 2 | 14.33<br>6.39<br>9.97<br>4.86<br>−0.84<br>1.71<br>4.60 | -<br>-<br>-<br>-<br>-<br>-<br>- | 8.73<br>12.04<br>40.74<br>1.85<br>−1.59<br>−3.83<br>1.06 | 9.62<br>7.07<br>13.27<br>4.95<br>4.10<br>19.91<br>−10.41 | [29] |
| China | Sub-tropical monsoon | Clay | 8.6 | 11.77 | Humic acid<br>Gypsum<br>Humic acid + gypsum | 0.6<br>0.6<br>0.6 + 0.6 | Rice | 1 | 18.37<br>2.30<br>10.45 | -<br>-<br>- | −3.77<br>9.43<br>−26.42 | −6.20<br>19.36<br>27.25 | [30] |
| China | Temperate continental monsoon | Loam | 6.8 | 8.81 | Humic acid + controlled-release fertilizer | - | Maize | 1<br>2 | 4.72<br>11.10 | −2.47<br>−2.94 | −40.94<br>−40.40 | -<br>- | [31] |
| Australia | Humid sub-tropical | Clay | 7.8 | 15.00 | Chicken manure + conventional N application rate<br>Composted chicken manure +conventional N application rate<br>Chicken manure + reduced N application rate<br>Composted chicken manure + reduced N application rate | - | Green beans + sorghum + broccoli + lettuce | 1 | −4.55<br><br>0.00<br><br>−6.82<br><br>2.27 | 14.16<br><br>11.80<br><br>11.98<br><br>−1.81 | −23.53<br><br>8.82<br><br>−41.18<br><br>23.53 | -<br><br>-<br><br>-<br><br>- | [32] |

Table 2. Effect of biochar application on mitigating GHG emissions under field conditions.

| Country | Soil Properties | | | | Biochar Feedstock | Pyrolysis temp. (°C) | Application Rate (Mg ha$^{-1}$) | Cultivated Crop | Years of Study | Yield (%) Compared with Control | GHG emissions Rate (%) Compared with Control | | | Reference |
|---|---|---|---|---|---|---|---|---|---|---|---|---|---|---|
| | Texture | pH | EC ds m$^{-1}$ | OC g kg$^{-1}$ | | | | | | | CO$_2$ | N$_2$O | CH$_4$ | |
| Australia | Clay loam | 4.6 | - | 4.5 | Cattle feedlot waste | 550 | 10.0 | Ryegrass | 3 | - | 0.27 | −13.67 | - | [38] |
| China | loam | - | - | - | Wheat | 550 | 20<br>40 | Corn-wheat | 2 | -<br>- | 0.13<br>−5.40 | 20.79<br>31.54 | -<br>- | [26] |
| USA | Silt loam | 7.7 | 0.4 | 18 | Wood | 500 | 22.4 | Corn Silage | 1<br>2<br>3 | 28.1<br>32.37<br>41.62 | 6.25 | 25.67 | 13.61 | [39] |

**Table 2.** *Cont.*

| Country | Soil Properties | | | | Biochar Feedstock | Pyrolysis temp. (°C) | Application Rate (Mg ha$^{-1}$) | Cultivated Crop | Years of Study | Yield (%) Compared with Control | GHG emissions Rate (%) Compared with Control | | | Reference |
|---|---|---|---|---|---|---|---|---|---|---|---|---|---|---|
| | Texture | pH | EC ds m$^{-1}$ | OC g kg$^{-1}$ | | | | | | | $CO_2$ | $N_2O$ | $CH_4$ | |
| China | - | - | - | - | Wheat straw | 450 | 20.0 | Corn | 1 | - | −4.17 | 34.34 | - | [40] |
| | | | | | | | 40.0 | | 2 | | −7.99 | 33.62 | - | |
| Japan | - | 8.45 | - | 6.3 | Bamboo | 700–800 | 20.0 | Kabocha squash | 1 | - | −346.4 | −42.85 | - | [41] |
| | | | | | | | | Bok choy | 1 | −173.68 | −347.0 | −29.50 | - | |
| | - | 8.30 | - | 14.1 | | | | Kabocha squash | 1 | 3.76 | −108.45 | −163.59 | - | |
| | | | | | | | | Bok choy | 1 | −38.02 | −146.83 | −0.56 | - | |
| Australia | Loamy sand | 7.1 | 0.11 | 2.6 | Wood | 500 | 10.0 | Grass | 2 | - | 6.25 | 19.50 | - | [42] |
| | | | | | | | 20.0 | | | - | −9.38 | 1.58 | - | |
| | Sandy loam | 6.4 | 0.04 | 1.8 | | | 10.0 | | | - | −18.24 | 9.18 | - | |
| | | | | | | | 20.0 | | | - | 17.57 | 19.77 | 40.00 | |
| China | - | 6.04 | - | 20.1 | Wheat straw | 350–550 | 10 | Rice | 1 | 10.0 | −1.05 | 7.14 | −14.00 | [43] |
| | | | | | | | 20 | | | 25.1 | 15.81 | 30.71 | −11.33 | |
| | | | | | | | 40 | | | 26.3 | 23.91 | 48.57 | −30.67 | |
| China | - | 8.1 | - | 49.6 | Corn straw | 550 | 26.0 | Cabbage + carrot | 1 | - | −20.12 | 11.19 | - | [44] |
| | | | | | | | 64.0 | | | - | −59.98 | −18.33 | - | |
| | | | | | | | 128.0 | | | - | −87.21 | 5.48 | - | |
| Germany | - | 7.1 | - | - | Beech wood | 400 | 60.0 | Corn | 3 | - | 42.86 | 62.96 | - | [45] |
| USA | Clay loam | 6.1 | 0.17 | 20.1 | Pinewood | - | 10.0 | Corn + soybean | 1 | - | −13.04 | 20.72 | - | [46] |
| | | | | | | | | | 2 | - | 9.16 | 7.58 | 20.76 | |
| | | | | | | | | | 3 | - | 1.33 | 32.41 | - | |
| | | | | | Corn stover | - | 10.0 | Corn + soybean | 1 | - | −6.60 | 19.55 | - | |
| | | | | | | | | | 2 | - | 11.90 | 9.41 | 20.13 | |
| | | | | | | | | | 3 | - | −13.65 | 11.11 | - | |
| | | | | | Switchgrass | - | 10.0 | Corn + soybean | 1 | - | −19.90 | 15.67 | - | |
| | | | | | | | | | 2 | - | 5.64 | 5.84 | 2.39 | |
| | | | | | | | | | 3 | - | −17.94 | 28.93 | - | |
| South korea | Silt loam | 5.18 | 0.50 | 17.8 | South korea Barley straw | 400 | | Chinese cabbage | 1 | - | - | 60.60 | - | [47] |
| China | Sandy loam | 6.5 | - | 18.1 | China Rice Straw | 600 | | Rice | Early paddy | 5.7 | 9.87 | 15.75 | −7.37 | [28] |
| | | | | | | | | | Late paddy | 3.6 | 25.36 | 43.60 | 15.07 | |
| China | - | 7.4 | - | 12.7 | Rice straw | 600 | 20.0 | Rice | 1 | 9.4 | - | −65.46 | 29.67 | [48] |
| | | | | | | | | | | 15.9 | - | 11.64 | 6.35 | |
| | | | | | | | 40.0 | | 2 | 24.0 | - | 58.01 | 29.87 | |
| | | | | | | | | | | 36.3 | - | 43.12 | 15.58 | |
| China | Sandy loam | 8.5 | 0.32 | 14.5 | Corn straw | 400 | 15.0 | Corn | 1 | 6.29 | 24.66 | 71.13 | - | [49] |
| | | | | | | | 30.0 | | | 7.34 | 17.57 | - | - | |
| | | | | | | | 45.0 | | | 1.77 | 22.13 | - | - | |
| | | | | | | | 15.0 | | 2 | 1.45 | 19.27 | 43.71 | - | |
| | | | | | | | 30.0 | | | 7.57 | 25.85 | 46.69 | - | |
| | | | | | | | 45.0 | | | 3.15 | 40.60 | 38.74 | - | |
| Canada | silty clay loam | 5.6 | - | 60.9 | Wheat straw | 450 | 20 | Barley | 1 | 20.37 | 6.22 | 33.18 | - | [50] |
| | | | | | | | | | 2 | | 3.56 | 34.32 | - | |
| China | - | 4.5 | - | 21.4 | - | - | 20.0 | Tea | 1 | 0.0 | - | 14.36 | - | [51] |
| | | | | | | | | | 2 | 0.0 | - | 7.10 | - | |

In temperate forests, proper management strategies (including biochar application) should be undertaken since about 70% of the global carbon fluxes are generated from respiration [52]. This modulating effect on the respiration of temperate forests might maximize the carbon pool of the soil matrix. In this regard, bamboo leaf biochar application at rates of 5 and 10 Mg ha$^{-1}$ increased carbon stock in moso bamboo forest by about 486 and 253%, respectively [53]. In the same regard, Ge et al. [54] recommended the combined application of biochar and nitrogen fertilizers for maintaining soil fertility and reducing $CO_2$ emissions from moso bamboo forest. The recalcitrant nature of biochar, its high water holding capacity, and its high potential to form soil aggregates with labile organic components may support its impact on maximizing soil carbon sequestration [53]. It is suggested that the safeguard effect of biochar against $CO_2$ emissions is associated with the predominance of soil bacterial species dealing with the tricarboxylic acid cycle [55]. Moreover, the key role of biochar in stimulating soil catalase, sucrose, urease, and β–glucosidase activities might support the protective effect of biochar against $CO_2$ emission from soil [56]. The stimulation of $CO_2$-fixing bacteria (e.g., *Chloroflexi*) following soil addition of biochar was also reported as a mechanism responsible for reducing $CO_2$ emissions [57].

On the other hand, cultivated land has almost lost 30–75% of its antecedent organic carbon pool as a result of soil respiration [58]. Therefore, effective soil management is highly recommended to maintain its soil carbon pool and offset the $CO_2$ emissions from cultivated land. To sustain the soil carbon pool in soil, organic additives application (e.g., farmyard manures) has been widely used as a common technique to sustain the productivity of soil. However, the high mineralization rate of organic carbon in soil (particularly under arid conditions) increases the potential of high $CO_2$ emissions from soil. Therefore, scientists recommend the application of organic manure in its pyrolyzed form given its carbon-negative nature. For instance, pig manure biochar application to sandy loam soil (northeast Toledo, Spain) decreased carbon mineralization compared to the original pig manure form [59]. In Zheng et al.'s work, biochar application over four years in sandy loam soil located in a semi-arid region (Henan Province, China) led to an increase in soil organic carbon (up to 22.1%), as well as the recalcitrant organic carbon fraction (up to 32.3%) [60]. Moreover, biochar application altered the bacterial community structure by motivating the abundance of *Chloroflexi* phylum (a bacterial community with low carbon mineralization and high carbon fixation potential).

In addition, biochar application was effective in reducing $CO_2$ emissions from saline soils in arid regions. Specifically, 400 °C corn stalks biochar application to coastal saline soil increased grain yield by about 28% without a significant effect on GHGs ($CO_2$, $CH_4$, and $N_2O$); however, the raw corn stalks increased $N_2O$ emission by 18% [61]. Conversely, other reports point to a neutral or stimulating effect of biochar in increasing $CO_2$ emission. For instance, the contribution of biochar additives (*Conocarpus erectus* L. at 400 °C) in mitigating $CO_2$ emissions from sandy–calcareous soil in Saudi Arabia was negligible relative to the original biomass form [62]. The stimulating effect of biochar application increasing $CO_2$ emissions could be due to the enhancement of dissolved organic carbon content, as well as the enrichment of *Proteobacteria* (copiotrophic bacteria) and inhibition of *Acidobacteriota* (oligotrophic bacteria) [63]. Besides, the role of biochar in reducing soil bulk density might provide a hospitable environment for microorganism respiration [56].

*3.2. Effect of Biochar Application on Reducing $N_2O$ Emissions*

Nitrous oxide ($N_2O$) is an atmospheric GHG derived mainly from agricultural practices (60%) that has greater global warming potential (300 folds) relative to $CO_2$ [64]. Scientists have noted a steady increase of $N_2O$ emissions of about 12% over the last 75 years (290 vs. 330 ppbv) [65]. Due to the progressive transition toward intensive agriculture, the excessive utilization of mineral fertilizers has led to the increase of $N_2O$ emissions by about 80% over the past century [66]. It is estimated that nearly 70% of global $N_2O$ emissions are generated from soil nitrogen transformation processes, which are associated with synthetic nitrogen inputs and soil tillage, including nitrification (conversion of $NH_4$

to $NO_3^-$) and denitrification (conversion of $NO_3^-$ to $N_2$) [67]. Interestingly, compost has been recognized as an important source for generating both of $CO_2$ and $N_2O$. The high $CO_2$ and $N_2O$ footprints of compost and other organic fertilizers cause substantial losses from the inherent carbon (0.1–10%) and nitrogen (2.0–3.0%) [68]. For example, livestock manure compost is produced annually by about $1.2 \times 10^6$ metric tons of $N_2O$ [69].

Biochar has been suggested as a non-sophisticated solution for maximizing the stability of organic fertilizers and reducing $N_2O$ emissions given its high porosity and high number of active functional groups. Data extracted from the literature point to a relatively low efficiency of biochar (24.64%) in mitigating $N_2O$ emissions (Figure 3).

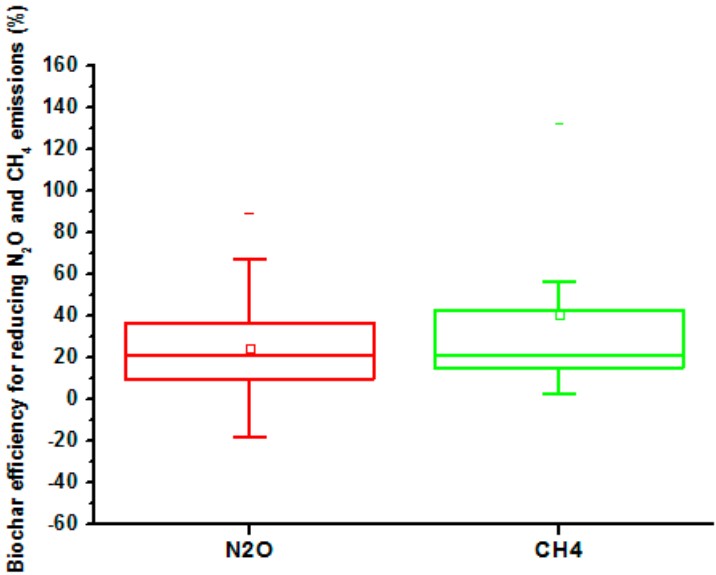

**Figure 3.** Biochar efficiency in reducing $N_2O$ and $CH_4$ emissions (%). Data are extracted from 19 investigations comprising 66 individual observations. Box chart is illustrated by the mean (dot), median (centerline), lower and upper quartiles (the lower and upper borders of the box, respectively), and whiskers-error bars (the minimum and maximum observations).

Authors have introduced numerous mechanisms that are responsible for alleviating $N_2O$ emissions. Song et al. suggested that biochar reduced $N_2O$ emissions directly by reducing the gross nitrification/denitrification rates of soil, and indirectly by reducing the content of soil-available N (ammonium, nitrate, and soluble organic nitrogen) and the activities of urease and protease [70]. Ji et al. [71] demonstrated that the application of biochar as a soil amendment was responsible for minimizing fungal abundance, thereby reducing $N_2O$ emissions by about 28%. The promotion of nitrifying bacteria and the inhibition of denitrifying bacteria by biochar application were also highlighted as mechanisms responsible for mitigating $N_2O$ emissions [72]. In addition, the inhibitory effect of biochar on bacterial-related nitrification/denitrification and N-cycle bacterial genes was further recorded in the rhizospheric layer of soil [73]. Another investigation pointed to the modulating effect of soil bulk density, nitrate reductase, nitrite reductase, and hydroxylamine reductase activities related to the denitrification process [74].

The ability of biochar to suppress $N_2O$ emissions has a long-lasting effect, even seven years following its soil application [75]. In a $^{15}N$-tracer incubation experiment, the protective effect of raw wood biochar (600 °C) on an alkaline soil (pH = 8.57) was due to the inhibition of the mineralization rate (gross autotrophic/heterotrophic nitrification and mineralization) and increasing the gross immobilization rates in soil [76]. The ability of biochar to modulate soil properties (e.g., aeration, pH/$E_H$, and organic carbon) has a subsequent effect on regulating soil nitrogen transformations and reducing $N_2O$ emissions [77].

Biochar soil application along with mineral nitrogen fertilizers might have a protective effect in reducing cumulative $N_2O$ emissions. In a microplot experiment, biochar appli-

cation (12 Mg ha$^{-1}$ of maize straw @ 450 °C) with $^{15}$N-labeled urea to sandy loam soil for three years maintained the retention of mineral nitrogen in the rhizosphere through reducing $N_2O$ emissions and $NO_3^-$ leaching [78]. Moreover, the combined application of biochar with nitrification inhibitors shows a high potential to reduce $N_2O$ emissions. In this regard, the application of manure biochar and nitrification inhibitor (nitrapyrin) reduced $N_2O$ emissions from urea by about 45.2% in a 60-day laboratory incubation experiment [79].

In contrast, other reports have suggested an adverse effect of biochar on increasing $N_2O$ emissions. For example, application of peanut shell biochar (pyrolyzed at 550 °C) along with nitrogen fertilizers to a grazing grassland (sandy loam in texture) in Queensland, Australia, increased $N_2O$ emissions compared to the unamended treatment (without biochar). This is mainly due to the increment of soil pH, which caused an abundance of narG, nirS, and AOB genes in the soil [80]. In this regard, the volatile matter of biochar has a key role in controlling the capacity of biochar for reducing $N_2O$ emissions since volatile matter content acts as a decomposable source of organic carbon for the denitrifying organisms [81]. Fresh biochar application was also responsible for increasing $N_2O$ emissions given the stimulation of the AOB-*amoA* gene abundance through autotrophic nitrification and denitrification [82].

### 3.3. Effect of Biochar Application on Reducing CH$_4$ Emissions from Rice Basins

Rice is the second largest cereal crop grown worldwide, representing the staple diet of two-thirds of the human population. A large amount (about 95%) of the world's rice yield is produced in developing countries, since it is considered an important source of employment and high income in rural areas. Among all agricultural ecosystems, paddy rice basins are one of the major sources of $CH_4$ emissions. The annual $CH_4$ emissions from paddy rice basins range between 31 and 112 Tg per year, which contributes about 5–19% of total greenhouse gas emissions [83]. Due to intensive wet rice farming all over the world, a tremendous increase in $CH_4$ emissions have been recorded by about 1.2 Tg/decade between 1961 and 2016 [84]. $CH_4$ is generated during the decomposition of organic matter by the aid of methanogenic archaea (methanogens) [85]. Although the contribution of $CH_4$ to total GHG emissions is not significant, it has a 25-fold higher global warming potential than $CO_2$ [86]. In addition, under waterlogging paddy ecosystems, the denitrification process is always very active and tends to convert nitrate to nitrous oxide [87].

The contribution of biochar to the reduction of $CH_4$ emissions has been highlighted in the literature. Analysis of recent literature showed moderate removal efficiency (40.49%) of $CH_4$ by biochar application to rice basins (Figure 3). The motivating effect of biochar application on activities of Acetyl-CoA synthase and β-glucosidase involved in carbon fixation reduced coenzyme activities related to methanogens [88]. Sriphirom et al. illustrated that 500 °C *Rhizophora apiculata* biochar application up to 4% led to a reduction in $CH_4$ emissions (9–21%) due to the abundance of electron donors (organic acids) and acceptors ($NO_3^-$, $SO_4^{2-}$, and $Fe^{3+}$), which accelerate redumethanogenesis reduction [89]. The beneficial effect of biochar in improving soil aeration and the readiness of $O_2$ supplies may inhibit methanogenesis [90]. In Wang et al.'s study, application of straw-derived biochar at 24 and 48 Mg ha$^{-1}$ mitigated $CH_4$ emissions by 20–51% through inhibiting the abundances of some methanogen populations (e.g., *Methanosaeta* and *Methanoregula*) [84]. Further study demonstrated that amending paddy soil with 550 °C biochar derived from *Rosa anemoniflora* branches inhibited the emission of $CH_4$ by motivating an abundance of mcrA and a high ratio of pmoA/mcrA [91]. The large surface area of biochar might also favor the electron transfer between bacteria and Fe minerals, thereby motivating the domination of Fe-reducing bacteria that discourage methanogens and inhibit $CH_4$ emissions [92]. The porous nature of biochar might promote $CH_4$ oxidation after adsorption into the abundant pores [93]. The biochar aging process might also facilitate the interaction with soil organic matter, Fe/Al oxides, and silicon, thereby forming coating layers on the internal and external surfaces of biochar [94].

The long-lasting effect of biochar application showed a high effectiveness for reducing $CH_4$ emissions. In the study of Nan et al., biochar derived from rice straw reduced the emissions of $CH_4$ over three successive years by about 43, 31, and 30%, and increased rice productivity by 8.0%, 1.6%, and 7.3%, respectively [95]. This long–term retardation of $CH_4$ emissions highlights the safeguarding effect of biochar as a stable and suitable microenvironment for carbon sequestration in rice basins. According to Wang et al., biochar application (24–48 Mg ha$^{-1}$) into rice paddies inhibited $CH_4$ emissions by 20–51% over four years of rice cropping due to the aerating effect of biochar, which enhanced the abundance of methanotrophic bacteria and decreased the abundance ratio of methanogens to methanotrophs [96].

In contrast, some reports have pointed to an increment of cumulative $CH_4$ emissions due to the abundance of methanogenic and methanotrophic genes in soil following biochar application [97]. The safeguarding effect of biochar against $CH_4$ emissions depends upon the pyrolysis technique and its temperature. Biochar showed higher efficacy for methane oxidation as compared with hydrochar, with higher pyrolysis temperature being superior. According to Liu et al., hydrochar application suppressed the growth of *Bacillus*, *Methylocystis,* and *Methylobacter*; however, biochar motivated an abundance of methane-oxidizing bacteria (methanotrophs) such as *Methylobacter* and *Methylocystis* [98].

## 4. Factors Affecting the Efficiency of Biochar in Mitigating GHG Emissions

### 4.1. Effect of Feedstock Type

The efficiency of biochar in reducing GHGs from soil depends mainly upon the inherent components of the feedstock. From recent published data (147 observations comprising 34 investigations), it is found that the average values of biochar efficiency in mitigating GHGs were 9.81, 6.68, and 20.37% with feedstocks of agricultural residuals, woody materials, and manures, respectively (Figure 4).

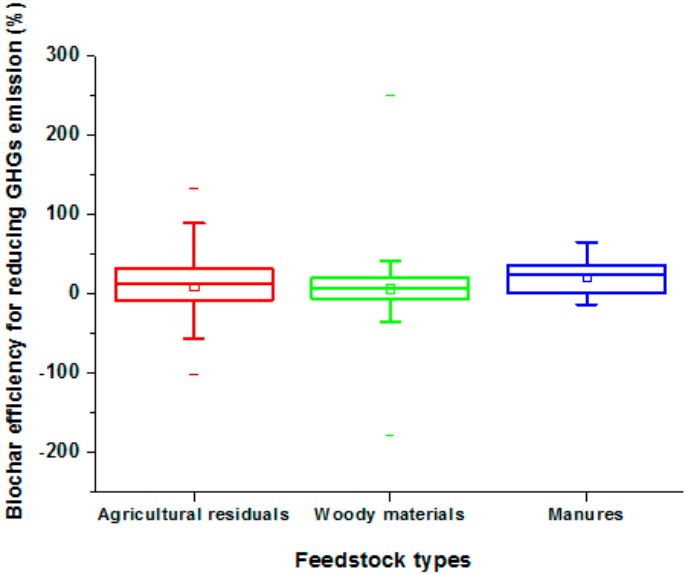

**Figure 4.** Effect of feedstock type on biochar efficiency in reducing GHG emissions from soil. Data are extracted from 34 investigations comprising 147 individual observations. Box chart is illustrated by the mean (dot), median (centerline), lower and upper quartiles (the lower and upper borders of the box, respectively), and whiskers-error bars (the minimum and maximum observations).

In other global meta-analyses, the magnitudes of $N_2O$ reduction index of biochars were feedstock-dependent: bamboo (31.9%) > field crop straw (27.1%) > manures (27.0%) > hardwood (18.1%) > field crop husks (0.47%) [99]. Contradictory results, however, were reported by other investigations since the high nitrogen content in manure-derived biochars favors its higher $N_2O$ emission footprint relative to other woody/herbaceous-derived

biochars [100]. Another study showed that plant-derived biochars showed higher values of aromatic carbon with high stability and resistance against microbial decay than other manure-derived biochars [101]. In a further study, there were no significant differences between manure and sawdust biochar (2.4 kg m$^{-2}$) for inhibiting emissions of $N_2O$ and $CH_4$ from soil [102]. It was also noticed that soils amended with biochars with a small C:N ratio exhibited higher $CO_2$ efflux than those amended with other biochars with a large C:N ratio [103].

*4.2. Effect of Pyrolysis Temperature*

Pyrolysis temperature plays a pivotal role in regulating GHG emissions following biochar soil application. The slow pyrolysis technique involves using low temperature, low heating rates, and high residence times to generate a high yield of high-quality biochar [104]. However, other pyrolysis types (fast and flash pyrolysis) generate low biochar yield with low surface functionality [105]. Specifically, the low rate of heating (24 °C min$^{-1}$) can form biochars with high aromaticity relative to the heating rate (62 °C min$^{-1}$) [106]. In terms of biochar stability, a higher mineralization rate was noticed with biochars produced under shorter residence time of pyrolysis compared to slow-pyrolyzed biochars, which exhibit less mobile organic matter and high resistance against microbial degradation [107]. The carrier gas (e.g., $N_2$, $CO_2$, or Ar) pointedly impact the yield, active functional groups, and volatile organic carbon content of biochar [108]. Moreover, the reactor design significantly affects the physicochemical properties of biochar. In this regard, a microwave pyrolysis reactor produces highly stable biochar relative to fixed/fluidized bed, rotating cone, screw feeder/auger, and vacuum pyrolizers [109]

Data extracted from recent literature illustrated that pyrolysis temperature was crucial for the performance of biochar in mitigating GHG emissions (Figure 5). Average values of inhibition efficiency of biochar-amended soils relative to the unamended ones are–60.71, 2.82, 25.42, and 7.86% with pyrolysis temperatures of 300–399, 400–499, 500–599, and 600–699 °C, respectively. Therefore, the utilization of low-pyrolyzed biochar in mitigating GHG emissions is not recommended given its high carbon footprint.

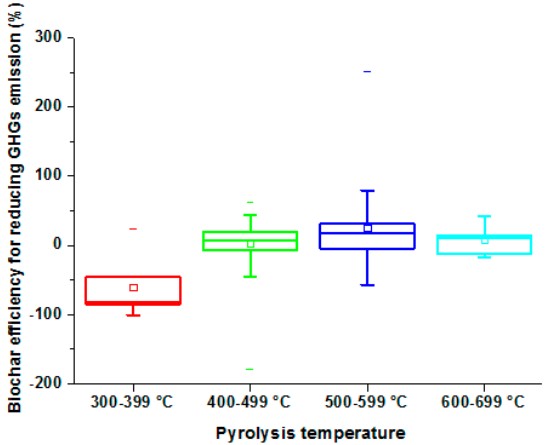

**Figure 5.** Effect of pyrolysis temperature on biochar efficiency in reducing GHG emissions from soil. Data are extracted from 26 investigations comprising 82 individual observations. Box chart is illustrated by the mean (dot), median (centerline), lower and upper quartiles (the lower and upper borders of the box, respectively), and whiskers-error bars (the minimum and maximum observations).

Based on a global meta-analysis, biochars produced at higher pyrolysis temperatures (>500 °C) exhibited higher potentials in reducing GHG emissions due to: (i) higher values of specific surface area, ash content, and polycondensed moieties; (ii) lower values of dissolved organic carbon, aliphatic compounds, and total surface charge; and (iii) suppressing the activity of soil microorganisms [110]. To summarize, low pyrolysis temperatures generate biochars with high volatile matter contents, low aromaticity, and high O:C ratio that are

less stable than those generated at high pyrolysis temperatures [111,112]. In the study of Yang et al., 300 °C maize straw biochar increased $CO_2$ emissions over the control treatment by about 46% owing to the increment of dissolved organic matter following the stimulation of copiotrophic bacteria (*Proteobacteria*) and the inhibition of oligotrophic bacteria (*Acidobacteriota*) [84]. In their study, they also reported that increasing pyrolysis temperature up to 450 and 600 °C reduced $CO_2$ emissions by about 10.5 and 14.0%, respectively, due to a subsequent decline in dissolved organic matter following biochar application. In Spain, biochar application (pig manure at 300 °C) to sandy loam soil resulted in a positive impact on dehydrogenase enzyme activity; however, the pyrolysis temperature of 500 °C did not show positive impacts on the activities of soil enzymes [59]. This modulating effect of biochar on soil enzymes was significantly correlated with $CO_2$ emissions by soil. According to Al-Rabaiai et al., high amounts of water-soluble organic compounds in biochars derived at low pyrolysis temperature might have a priming effect on stimulating microbial activities and soil respiration [85]. In contrast, spent-mushroom-substrate-derived biochar applied to moso bamboo forest soil at the rate of 50 g kg$^{-1}$ caused a considerable increase in $CO_2$ emissions by about 73, 43, and 16.6% with pyrolysis temperatures of 300, 450, and 600 °C, respectively [113].

Likewise, the performance of biochar in mitigating $N_2O$ emissions declined sharply at low pyrolysis temperatures owing to the smaller surface area and the low aromaticity of the produced biochar [114]. In addition, the extremely high pyrolysis temperature (900 °C) led to an increase in $N_2O$ emissions from soil amended with walnut shell biochar due to favoring the denitrification process [115]. It was also reported that a high pyrolysis temperature of biochar (700–900 °C) can increase the cumulative emission of $CH_4$ fluxes, given the formation of biochars with highly condensed aromatic graphite structures, motivating electronic transfer ability of methanogens and the inherent salts in biochar additives [116,117]. In an incubation experiment, $CH_4$ generation from a paddy soil following straw biochar application was ranked as follows: biochar at 300 °C > biochar at 500 °C > biochar at 700 °C [118].

### 4.3. Effect of Application Rate

Restraining GHG emissions by biochar is an application-dependent strategy. Biochar often applied through broadcasting and mixing with soil matrix by tillage methods [58]. However, this application method is responsible for wind loss of biochar by about 25% of the applied amount [119]. In addition, biochar is frequently applied to a trench as furrow application after crop planting with a lower amount of application and minimal soil disturbance [120]. On the other hand, biochar could be indirectly applied to the soil by mixing with several soil amendments (e.g., lime, compost, and manure) [121,122].

A subsequent initial flush of $CO_2$ is emitted from soil following biochar application, which declines sharply with the recalcitrant aged biochar [123]. The inhibitory effect of biochar application to $CO_2$ emissions can be associated with the sorption of rhizodeposits and enzymes onto biochar, which lead to a reduction in carbon-degrading microbial activity in soil [124]. Numerous attempts have been made to specify the optimum rate of biochar soil application; however, a great deal of uncertainty remains surrounding the appropriate rate for each soil type. The extracted data from 31 investigations comprising 146 individual observations illustrate the average removal efficiency values of GHGs from biochar-amended soils relative to the unamended ones. Biochar efficiency in reducing GHG emissions was, on average, 15.91, 6.13, −2.13, and −4.45% with application rates of 1–10, 11–20, 21–40, and >40 Mg ha$^{-1}$ (Figure 6). According to the obtained results, there seems to be a consensus that the low rate of biochar (up to 10 Mg ha$^{-1}$) is more preferable to achieve net carbon neutrality. However, a high application rate of biochar might increase the cumulative GHG emissions.

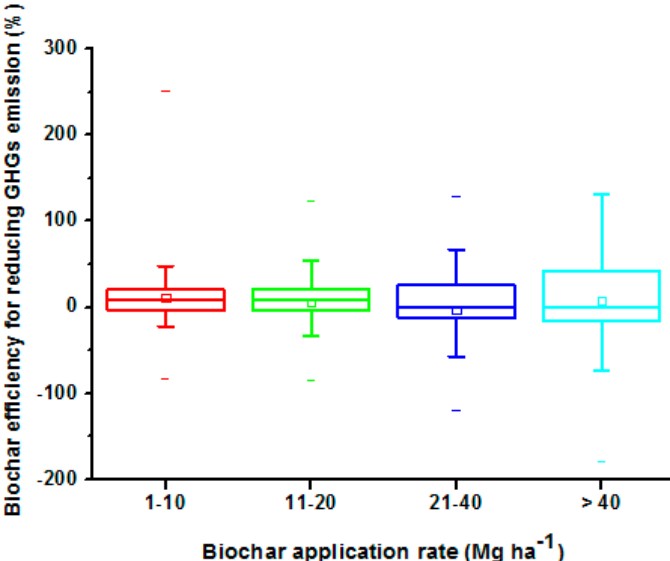

**Figure 6.** Effect of biochar application rate on efficiency in reducing GHG emissions. Data are extracted from 26 investigations comprising 82 individual observations. Box chart is illustrated by the mean (dot), median (centerline), lower and upper quartiles (the lower and upper borders of the box, respectively), and whiskers-error bars (the minimum and maximum observations).

Former studies have monitored the effect of the biochar application rate on the efficacy of regulating GHG emissions. Under upland rice production, the highest application rate of 350 °C rice-husk-derived biochar (25 Mg ha$^{-1}$) showed the highest $CO_2$ emissions (3.06 $CO_2$–C g/m$^2$) relative to the one-fifth application rate (2.78 $CO_2$–C g/m$^2$) [125]. Similarly, a low level of biochar (5 Mg ha$^{-1}$ of 500 °C bamboo branches) could be more effective than the high application dosage (20 Mg ha$^{-1}$) in reducing $CO_2$ emissions and improving soil carbon sequestration in forest soils [126]. Another investigation revealed that increasing the application rate of bamboo leaf biochar did not show substantial alterations in $CO_2$ emission [127].

As mentioned earlier, the optimum application rate of biochar might change according to the soil type. For instance, soil biochar application (20 Mg ha$^{-1}$ 650 °C) to deciduous mixed forest led to a substantial decline in $CO_2$ emissions; however, this emission rate was not significantly changed in a long-term fertilized apple orchard [128]. Other reports suggested raising the application rate of biochar to offset the cumulative GHG emissions. In the study of Shen et al., increasing the application rate of 450 °C maize straw biochar from 10 up to 30 Mg ha$^{-1}$ was associated with a progressive reduction in cumulative $CO_2$ emissions (from 3.9 to 11.8%) due to the sorption of labile carbon onto internal and external surfaces of biochar, thus suppressing the rate of soil respiration [129].

The application rate also affects the performance of biochar in regulating GHG emissions from different soil layers. For instance, cumulative $N_2O$ emissions from the topsoil (0–5 cm) following the application of pruning waste biochar (@ 600 °C) at rates of 2 and 10% declined $N_2O$ emissions by 12.5% and 26.3%, respectively. However, the safeguard effect of pruning waste biochar in reducing $N_2O$ emissions from a soil layer of 0–10 and the rhizospheric layer was only observed with the rate of 10% (15.1 and 13.8%, respectively) [130]. Biochar application as film-mulching has recently been investigated on cropping systems grown under drip irrigation. In view of that, corn-residue-derived biochar (produced under pyrolysis at 400–500 °C) was applied as a film mulch to drip-irrigated corn grown in sandy loam soil in Inner Mongolia, China. Results showed that increasing the application rate from 15 up to 45 Mg ha$^{-1}$ as a film mulch system was associated with significant reductions in GHGs over two growing seasons: $CO_2$ (19 –33%), $CH_4$ (124–132%), and $N_2O$ (55–79%) [49].

## 5. Effect of Biochar on Reducing GHG Emissions during Composting

Composting is a process in which organic wastes are transformed via complicated biochemical reactions into recalcitrant organic products (humic substances in particular) that can serve as fertilizers to sustain soil fertility and productivity [131]. However, the composting process is responsible for emitting substantial amounts of GHGs that raised concerns from ecological point of view. In particular, $CH_4$ accounts for about 6% of the total carbon loss during the composting process, and the emission of $N_2O$ accounts for approximately 3.8% of the total nitrogen losses [132]. Furthermore, the release of ammonia ($NH_3$) during composting not only harms the ecosphere but also declines the agricultural revenues from compost additives. Related research investigations revealed that animal husbandry is the main source of agricultural non-$CO_2$ emissions, accounting for about 37 and 65% of $CH_4$ and $N_2O$, respectively (12% of the anthropogenic GHG emissions globally) [133]. The emission of GHGs is more pronounced in the initial phase of the composting process (the thermophilic phase), in which the temperature reaches about 70 °C; however, this emission declines dramatically in the ultimate maturation phase (the mesophilic phase), when the temperature drops to 40–50 °C [134].

The applicability of biochar additives to reducing GHG emissions during composting is illustrated in Table 3. The high functionality of biochar has attracted research attention for utilization as a supplemental additive during the composting process to accelerate the startup of decomposition, shorten the period of composting, and reduce the amounts of GHG emissions [135]. Furthermore, biochar might reduce the mobility of water-soluble organic substances and avoid their losses during the composting process [136]. In view of this, biochar application during composting might inhibit the abundance of the nirK gene in denitrifying bacteria, which causes a significant reduction of $N_2O$ emissions during composting [137]. In the study of Guo et al., bamboo charcoal application during composting led to minimizing $NH_3$ emissions following the active nitrification by *Nitromonas* [138]. In another study to evaluate the effect of 5% biochar application during pig manure composting, it was reported that biochar could shorten the composting period and reduce emissions of $CO_2$, $CH_4$, $N_2O$, and $NH_3$ by about 35.9, 15.4, 19.9, and 18.8%, respectively [139]. However, in another study, chicken manure biochar application (up to 10%) during chicken manure composting declined the release of GHGs: $N_2O$ (19.0–27.4%), $CH_4$ (9.3–55.9%), and $NH_3$ (24.2–56.9%) [132]. Likewise, the co-application of bamboo biochar with poultry manure during composting (up to 10% w:w) reduced $CO_2$ and $NH_3$ losses by about 542–149% and 48–11%, respectively [140].

The composting of sewage sludge is a green technology for reducing the negative impacts associated with its ecological hazards [141]. However, a great deal of uncertainty arises surrounding its high GHG footprint [142]. To address this environmental constraint, Awasthi et al. [143] added rice straw biochar at a high rate (8–18%) during sewage sludge composting and recorded remarkable declines in GHG emissions: $CH_4$ (93–95%), $NH_3$ (58–65%), and $N_2O$ (95–97%). In yet another investigation, bamboo biochar application along with bacterial agents during sewage sludge composting mitigated $CH_4$ and $N_2O$ emissions (45.7% and 3.7%, respectively) due to the beneficial effect of biochar on filling the space between compost particles, thereby minimizing the potential heat losses and motivating microbial activity and consequent heat production [144]. Vermicomposting is a benign and modern eco-friendly technique for addressing the vast accumulation of organic wastes, although the incorporation of earthworms in these organic fertilizers showed higher emissions of $N_2O$ relative to other traditional composts [145]. According to Wu et al., the incorporation of 500 °C rice straw biochar with vermicompost significantly reduced cumulative $N_2O$ emissions (~19%) [146].

The application of biochar as a ruminant diet showed a significant impact on improving the digestion performance of animals, reducing enteric $CH_4$ and increasing the value of post-excretion biochar–manure mixture. In a recent study, the effect of pine-based biochar application on cattle diet increased stockpile/compost aromaticity with a high content of more humic-like organic matter precursors [147].

**Table 3.** Effect of biochar application on mitigating GHG emissions during composting.

| Feedstock | Biochar Characteristics | | | Composting Materials | Compost Characteristics | | GHG Emissions Reduction Effect (% Compared with Control) | | | | References |
|---|---|---|---|---|---|---|---|---|---|---|---|
| | Pyrolysis Temperature (°C) | Biochar Application (% *w/w*) | Particle Size (mm) | | Bulk Density (g cm$^{-3}$) | C/N Ratio | $CO_2$ | $CH_4$ | $N_2O$ | $NH_3$ | |
| Hardwood+ softwood | 500–700 | 27.4 | ≤16 | Hen manure + barley straw | 0.49 | 17.4–13.7 | 21.5–22.9 | 77.9–83.6 | 35.3–43.0 | 35.3–43.0 | [148] |
| Woody material | - | 4 | - | Sewage sludge + woodchips | - | 22–25 | - | - | - | 8.5–9.2 | [149] |
| Holm oak | 650 | 10 | - | Green waste + municipal solid waste | - | 27.5–16.2 | 52.9 | 95.1 | 14.2 | - | [150] |
| Green waste + poultry litter | 550 | 10 | - | Poultry litter + sugarcane straw | - | - | - | 77.8–83.3 | 68.2–74.9 | 54.9 −60.2 | [151] |
| Wheat straw | 500–600 | 2–18 | 2–5 | Sewage sludge + wheat straw | 0.50 | 25.0 | - | 92.8–95.3 | 95.1–97.3 | 58.0–65.2 | [143] |
| Bamboo | - | 5 | 2–3 | Pig manure + sawdust | 0.50 | - | - | 54.4 | 36.1 | 12.4 | [152] |
| Chicken manure + wheat straw | 550–600 | 2–10 | - | Chicken manure | 0.50 | 16.8–14.2 | - | 24.4–63.4 | 6.8–16.9 | 22.9–50.5 | [132] |
| Bamboo | - | 2–10 | - | Poultry manure + wheat straw | 0.50 | 24.3–18.9 | 5.5–72.6 | 12.5–72.9 | 12.4–81.6 | 19.0–77.4 | [140] |
| Cornstalk | 450–500 | 10.0 | ≤2.0 | Hen manure, sawdust | 0.42 | - | - | - | - | 12.4 | [153] |
| Waste wood pellets | 520 | 10 / 15 | - | Chicken mortalities | - | 20.3 / 20.0 | - / - | - / - | - / - | 40.0 / 56.8 | [154] |

## 6. Designer/Functionalized Biochar for Efficient Retardation of GHG Emissions from Soil and Terrestrial Ecosystems

The scientific community is interested in maximizing the functionality of biochar so that it can be tailored to a variety of agro-environmental applications [11–13,155]. In a meta-analysis study, acidic oxidation was the most efficient method for enhancing the physicochemical properties of biochar (specific surface area, micro-pores, oxygen-containing functional groups, and cation exchange capacity) relative to other oxidation methods (physical, alkaline, metal oxide, and natural oxidation methods) [156]. Recently, designer biochar functionalization (specialization) has attracted increasing attention in recent research to lessen GHG emissions from agricultural ecosystems. Several reports highlighted the activation of pristine biochar (raw biochar) via physical and chemical modification methods to improve the functionality of biochar (increase its surface area and porous structure) for the sorption of GHGs. However, most of these reports are still under lab-scale experimentations. In addition, most of these reports are focused on capturing $CO_2$ emissions. Data extracted from published reports (97 values) showed that the average values of $CO_2$ sorption with pristine, physically activated, and chemically activated biochars were 37.8, 56.5, and 59.4 mg $g^{-1}$, respectively (Figure 7).

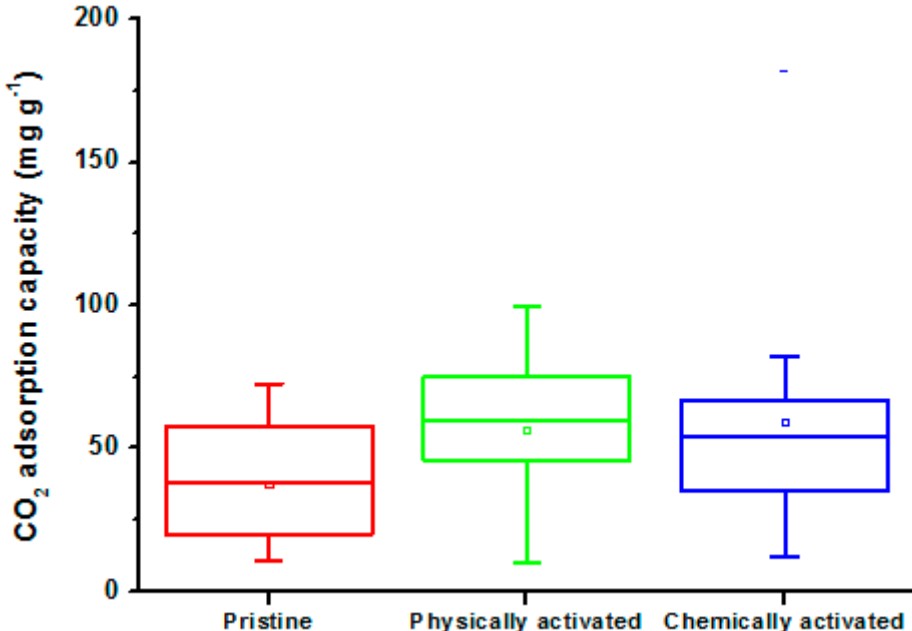

**Figure 7.** Effect of activation methods on sorption capacity of $CO_2$ (mg $g^{-1}$) with pristine and designer biochars. Data are extracted from 97 individual observations. Box chart is illustrated by the mean (dot), median (centerline), lower and upper quartiles (the lower and upper borders of the box, respectively), and whiskers-error bars (the minimum and maximum observations).

A wide range of researchers have highlighted physical modification as the preferable activation method to maximize the functionality of biochar relative to chemical modifications methods [157,158]. Proponents of physical modification point to the high risk of deteriorating the carbonaceous lattice following chemical modifications and the potential to block the pore structures of the biochar matrix, which might reduce gases sorptivity [159]. Steam and high-temperature gas activation have been attempted to improve the porosity of biochar; however, the $NH_3$ activation not only improved the pore structure of activated biochar, but also introduced active functional groups onto the carbonaceous lattice [160]. Physical treatment with $NH_3$ can also increase the alkaline nature of biochar and the base–acid interaction between $CO_2$ and the originated nitrogen-containing functional groups [161]. Furthermore, $NH_3$ treatment grafted pyrrolic–N groups onto the biochar

matrix, which facilitated the H–bonding interaction between $CO_2$ molecules, and the proton of pyrrolic–N [162].

Opponents of physical activation methods, however, point toward the higher energy consumption, longer activation time, and higher activation temperature. Several chemical activation methods have been reviewed for the functionalization of biochar for capturing GHGs. Among them, alkali and acid-modified biochars have shown high sorptivity with low cost and simple processing. Alkali treatment of wood pellet biochar maximized the capture capacity of $CO_2$ by about five folds relative to untreated biochar (50.73 vs. 10.45 mg $g^{-1}$) due to improving surface area, porosity, and abundance of active functional groups on the biochar matrix [163]. High $CO_2$ capture (160 mg $g^{-1}$) was further recorded by 350 °C pine cone biochar activated with KOH [164]. In yet another study, high capacities of $CO_2$ retardation (136.7–182.0 mg $g^{-1}$) were recorded by KOH-activated biochars derived from pine sawdust and sewage sludge mixture as compared with plain biochar (35.5–42.9 mg $g^{-1}$) given the formation of tunable porous features in biochar matrix [165].

In their studies on rice plant, Shin and coworkers [166] reported that activated biochar (alkali-treated rice hull pellets)-doped mineral fertilizer (40% N) reduced cumulative $CO_2$ and $N_2O$ emissions by about 10 and 0.003 kg $ha^{-1}$, respectively, compared to the control treatment, with a negligible effect on the emission of $CH_4$. In another experiment, 700 °C rice husk biochar was acid modified with $H_3PO_4$ and further combined with nano-zero-valent iron (nZVI) to enhance its sorption capacity for GHGs. The functionalized biochar form reduced $CO_2$ and $N_2O$ emissions; however, $CH_4$ emission showed a noticeable increase [167].

The doping of heteroatoms (e.g., Mg, N, S, and Ni) into the carbonaceous lattice of biochar exhibited promising values for capturing GHGs due to the electron-withdrawal effect. In view of this, Mg-doped rice straw biochar application (at 9%) to a dryland soil in Hunan Province, China, showed a minimal effect on $CH_4$ emission but reduced cumulative emissions of $CO_2$ (9%) and $N_2O$ (32%) as compared to the control treatment [168]. Nitrogen-doped biochar (mixture of rice straw and waste wood pyrolyzed at 600 °C and applied at 0–8 Mg $ha^{-1}$) was further studied in rice cropping soils under a short-term study [169]. Compared with the control treatment, nitrogen-doped biochar application increased $CO_2$ emissions and reduced emissions of $CH_4$ with the application rate of 8 Mg $ha^{-1}$.

Iron species receive high attention for biochar specialization in several agro-environmental applications. As a result of its effect on denitrification functional genes (nirk, narG, nirS, and nosZ), methanogenesis (mcrA), and methanotrophs (pmoA), modified biochar (derived from reed, walnut, saw dust, and sludge) supported by nano-zero-valent iron (nZVI) was able to reduce $N_2O$, $CO_2$, and $CH_4$ emissions [170]. Biochar (600 °C *Camellia oleifera* fruit shell) modified with $Fe(NO_3)_3$/KOH showed a high efficacy in $N_2O$ retardation with an increment of about 8.6% over pristine biochar [171]. Developing enriched biochar derivatives with higher functionality (organo-mineral complexes, surface area, exchanging capacity of ions and dissolved organic carbons) has been proposed to enhance the sequestration of GHGs. The carbon footprint of either conventional rice straw biochar or enriched biochar with lime, clay, ash, and manure was compared under paddy field conditions [172]. They indicated that the difference in the carbon footprint between biochar types is mainly associated with variations in $CH_4$ emissions among plain and functionalized biochars.

## 7. Conclusions and Future Prospects

In recent years, lowering the rise in emissions of greenhouse gases has become one of the issues that requires global attention. In this work, we performed a review of the recent literature on biochar, as a tool to lessen the impact of emissions and mitigate their negative consequences. Overall, this review assesses the different methods of biochar production and their effectiveness. This review primarily summarizes the biochemical processes occurring in the charosphere. In addition, recent developments in our understanding of how to activate biochar for maximum effectiveness in achieving carbon neutrality goals are covered. To conclude:

- Biochar and its derivatives have been shown to reduce emissions of $CO_2$, $N_2O$, and $CH_4$ from soil and organic manures. Recent field-scale studies have found that biochar has the potential to reduce emissions of $CO_2$, $N_2O$, and $CH_4$ on the order of $-0.430$, 24.681, and 40.486%, respectively.
- While biochar showed promising results in reducing greenhouse gas emissions when tested in controlled laboratory settings, studies conducted on a larger scale have shown either no effect or even a negative one on efforts to lower GHG emissions.
- Notably, biochar's effectiveness in cutting greenhouse gas emissions is proportional to its application rate, pyrolysis temperature, and the type of feedstock used to make it.
- To produce effective amendments with a high capacity for restraining GHG production and enhancing soil carbon sink, it is recommended that manure-derived biochars be pyrolyzed between 500 and 600 $^\circ$C, and applied at a rate of less than 10 Mg ha$^{-1}$.
- Overall, biochar can be seen as a highly effective and relatively simple tool for reversing the upward trend in greenhouse gas emissions.
- Since carbon and nitrogen transformation processes are microbially dependent, future research should be directed toward (i) investigating other indirect factors related to soil physicochemical characters (such as soil pH/Eh, colloidal and $CaCO_3$ contents) that may control the functionality of biochar, (ii) fabricating aged biochars with low carbon and nitrogen footprints, and (iii) functionalizing biologically activated biochars to suppress $CO_2$, $CH_4$, and $N_2O$ emissions.

**Author Contributions:** Conceptualization, A.M. and M.M.M.; methodology, A.M. and E.S.; software, A.E.-G. and M.E.A.; validation, A.M., M.M.M. and A.M.E.K.; formal analysis, A.M. and E.S.; investigation, A.M. and M.M.M.; resources, M.E.A. and A.M.E.K.; data curation, A.M. and E.S.; writing—original draft preparation, A.M., M.M.M. and E.S.; writing—review and editing, A.M., M.E.A. and A.M.E.K.; visualization, A.E.-G. and E.S.; supervision, A.M. and A.E.-G.; project administration, A.M. and A.M.E.K.; funding acquisition, A.M.E.K. and M.E.A. All authors have read and agreed to the published version of the manuscript.

**Funding:** This research received no external funding.

**Institutional Review Board Statement:** We choose to exclude this statement because our study did not involve humans or animals.

**Informed Consent Statement:** We choose to exclude this statement because our study did not involve humans or animals.

**Data Availability Statement:** Not applicable.

**Acknowledgments:** The authors would like to acknowledge Mansoura University for supporting the publication fees of this manuscript.

**Conflicts of Interest:** The authors declare no conflict of interest.

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
