# Peer review of "Biochar as a Soil Amendment for Restraining Greenhouse Gases Emission and Improving Soil Carbon Sink: Current Situation and Ways Forward"

_sustainability, doi:10.3390/su15021206_

Round 1
Reviewer 1 Report
MS Number : sustainability-2033260
MS Title : Biochar as a soil amendment for restraining greenhouse gases emission and improving soil carbon sink in arid and semi-arid regions: Current situation and ways forward
Reducing the increase in greenhouse gas emissions in recent years is one of the issues that need global attention and attention. For this purpose, this study was planned in order to reduce this negative effect and minimize the effect of the emission. Therefore, an experiment was set up. The findings are important and it is appropriate to accept the article after minor corrections.
I think the paper is needed to support new references about biochar application and biochar characteristics ratio to see a new aspect. to improve, please support the introduction with these refs. Please incorporate all of the introductions about biochar and its effects.
The methods section is needed to support with latest references. Because some of the methods are references.
If possible, please change the figures in the paper.
The results section was perfect. However, Discussion is needed on more information on different biochar production such as pyrolysis technology and application methods to improve, please incorporate with the latest refs. Please incorporate all of them in the discussion section.
The conclusion is needed rephrases. Please extend and put in your final recommendation.
Author Response
Responses to Reviewers' comments
We appreciate the valuable comments from the reviewers, which improved the revised version of our manuscript. We provided a full consideration for these comments, and detailed amendments are listed below in our point-by-point response. Reviewer's comments are highlighted in a blue color, and the response to these comments is presented in a black color. In addition, changes are highlighted in a red-color style in the revised version of the manuscript.
Authors' responses to Reviewer 1's comments:
Comment 1: Reducing the increase in greenhouse gas emissions in recent years is one of the issues that need global attention. For this purpose, this study was planned in order to reduce this negative effect and minimize the effect of the emission. Therefore, an experiment was set up. The findings are important and it is appropriate to accept the article after minor corrections.
Response 1: Thank you for the positive comment and laud!
Comment 2: I think the paper is needed to support new references about biochar application and biochar characteristics ratio to see a new aspect to improve, please support the introduction with these refs. Please incorporate all of the introductions about biochar and its effects.
Response 2: Thank you for the positive comment. The introduction was supported by updated references illustrating the urgent need to functionalize modern biochars with high functionality toward restraining GHGs under field-scale applications. This promising research topic could provide a roadmap towards a sustainable future utilization of biochar as a soil amendment for restraining GHGs emissions (Lines 74-92).
The high functionality of biochar, including its physical properties (porosity, large surface area and high water holding capacity) as well as its chemical properties (abundance of oxygen–containing functional groups, surface charge and pH modulating effect), suggests its potential utilization in reducing GHGs emissions [15]. However, a great deal of uncertainty remains surrounding the competitiveness of biochar with traditional soil amendments (e.g compost and farmyard manures) given its low nutrient content and high pH value. Additionally, the efficacy of pristine biochar for restraining GHGs emissions under field-scale applications is still questionable. The wide range of variation between biochars depends upon the multiplicity of factors that underlying thermochemical conversion of organic biomass [16-18]. Several key factors are controlling the property of biochar and its effectiveness in reducing GHGs emission including (i) feedstock type (e.g. agricultural wastes, sludge and manures, algal biomass and crustacean shell wastes), (ii) pyrolysis type (slow/fast pyrolysis, gasification, torrefaction, and hydrothermal carbonization), (iii) thermal processing protocol (e.g. heating rate, pressure and carries gas, residence time and reactor design) and (iv) soil application rates and methods (e.g. broadcasting, in-furrow or mixture with soil amendments) [19]. Consequently, biochar functionalization has emerged as a new trend to provide a roadmap for enhancing the competitiveness of biochar and its sustainable soil application for reducing GHGs emissions.
Comment 3: The methods section is needed to support with latest references. Because some of the methods are references.
Response 3: We appreciate this valuable comment. As this is a review article, we have clarified the procedure for selecting the research articles, books, chapters, etc including in this review. (lines 622-649).
Books, book chapters, research articles, review articles, and proceedings were all scoured for this review. All selected sources were written in English and published within the last decade (2010 onwards). All of these articles came straight from reputable sources (e.g. Scopus, Web of Science, ProQuest, EBSCO and JSTOR). The International Biochar Initiative (IBI), the United States Department of Agriculture (USDA), and the European Environment Agency all contributed to the credibility of our review by providing official reports, statistics, and proceedings. Use of Get-Data Graph Digitizer (ver. 2.22, Russian Federation) allowed us to convert the data visualizations into corresponding numerical values. In this review, biochar + carbon dioxide emissions, biochar + nitrogen oxide emissions, biochar + chlorofluorocarbon emissions, biochar + soil + carbon dioxide emissions, biochar + soil + nitrogen oxide emissions, and biochar + soil + chlorofluorocarbon emissions were the initial search keywords. In addition, a number of meta-analyses were reported from a variety of published articles to establish a solid assessment of the current state of the potential use of biochar for limiting GHGs emissions and the prospects for this direction.
Comment 4: If possible, please change the figures in the paper.
Response 4: In order to provide a visual representation of our message, the use of figures is crucial. All figures have been updated to be clearer and higher quality per the reviewer's request.
Comment 5: The results section was perfect. However, discussion is needed on more information on different biochar production such as pyrolysis technology and application methods to improve, please incorporate with the latest refs. Please incorporate all of them in the discussion section.
Response 5: Many thanks for your comment. This review article summarized the prominent findings of published articles about utilization of biochar for restraining GHGs emissions. More discussion was provided in the revised version of manuscript regarding the effect of pyrolysis technology (Lines 339-353) and application methods (Lines 402-408) on regulating the functionality of biochar.
-Pyrolysis temperature exhibits a pivotal role in regulating GHGs emissions following biochar soil application. Slow pyrolysis technique involves using low temperature, low heating rates and high residence times to generate high yield of high-quality biochar [74]. However, other pyrolysis types (e.g. fast and flash pyrolysis) generate low biochar yield with low surface functionality [75]. Specifically, the low rate of heating (24 °C min−1) can form biochars with high aromaticity relative to high heating rate (62 °C min−1) [76]. In terms of biochar stability, a higher mineralization rate was noticed with biochars produced under shorter residence time of pyrolysis compared to slow-pyrolyzed biochars, which exhibit less mobile organic matter and high resistance against microbial degradation [77]. The carrier gas (e.g. N2, CO2 or Ar) pointedly impact the yield, active functional groups and volatile organic carbon content of biochar [78]. Moreover, the reactor design emphatically affects physicochemical properties of biochar. In this regard, microwave pyrolysis reactor produces highly stable biochar relative to fixed/ fluidized bed, rotating cone, screw feeder/auger and vacuum pyrolizers [79]
-Restraining GHGs emissions by biochar is an application-dependent strategy. Biochar often applied through broadcasting and mixing with soil matrix by tillage methods [89]. However, this application method is responsible for wind loss of biochar by about 25% of the applied amount [90]. In addition, biochar is freuently applied to a trench as furrow application after crop planting with less amount of application and minimal soil disturbance [91]. On the other hand, biochar could be indirectly applied to the soil by mixing with several soil amendments (e.g. lime, compost and manure) [92,93].
Comment 6: The conclusion is needed rephrases. Please extend and put in your final recommendation.
Response 6: Thank you for the suggestion. The conclusion was rephrased and the final recommendation is incorporated in the updated version of manuscript (lines 622-649).
“In recent years, lowering the rise in emissions of greenhouse gases has become one of the issues that needs global attention. In this work, we made a review of recent literature (2010-2020) on biochar, as a tool to lessen the impact of the emission and mitigate its negative consequences. Overall, this review assesses the different methods of biochar production and their effectiveness. This review primarily summarizes the biochemical processes occurring in the charosphere. In addition, recent developments in our understanding of how to activate biochar for maximum effectiveness in achieving carbon neutrality goals are covered. To conclude:
- Biochar and its derivatives have been shown to reduce emissions of CO2, N2O, and CH4 from soil and organic manures. Recent field-scale studies have found that biochar has the potential to reduce emissions of CO2, N2O, and CH4 on the order of -0.430, 24.681, and 40.486%, respectively
- While biochar showed promising results in reducing greenhouse gas emissionswhen tested in controlled laboratory settings, studies conducted on a larger scale have shown either no effect or even a negative one on efforts to lower GHG emissions.
- Notably, biochar's effectiveness in cutting greenhouse gas emissions is proportional to its application rate, pyrolysis temperature, and the type of feedstock used to make it.
- To produce effective amendments with a high capacity for restraining GHGs production and enhancing soil carbon sink, especially in arid and semi-arid regions, it is recommended that manure-derived biochars be pyrolyzed between 500 and 600 °C, and applied at a rate of less than 10 Mg ha-1.
- Overall, biochar can be seen as a highly effective and relatively simple tool for reversing the upward trend in greenhouse gas emissions.
- Since carbon and nitrogen transformation processes are microbially dependent, future research should be directed toward (i) investigating other indirect factors related to soil physicochemical characters (such as soil pH/EH, colloidal and CaCO3 contents) that may control the functionality of biochar, (ii) fabricating aged biochars with low carbon and nitrogen footprints, (iii) functionalizing biologically-activated biochars to suppress CO2, CH4 and N2O emissions, and (iv) fabricating.
- Zhu X, Labianca C, He M, Luo Z, Wu C, You S, et al. Life-cycle assessment of pyrolysis processes for sustainable production of biochar from agro-residues. Bioresource technology. 2022:127601.
- Law XN, Cheah WY, Chew KW, Ibrahim MF, Park Y-K, Ho S-H, et al. Microalgal-based biochar in wastewater remediation: Its synthesis, characterization and applications. Environmental research. 2022;204:111966.
- Güleç F, Williams O, Kostas ET, Samson A, Lester E. A comprehensive comparative study on the energy application of chars produced from different biomass feedstocks via hydrothermal conversion, pyrolysis, and torrefaction. Energy Conversion and Management. 2022;270:116260.
- Al-Rumaihi A, Shahbaz M, Mckay G, Mackey H, Al-Ansari T. A review of pyrolysis technologies and feedstock: A blending approach for plastic and biomass towards optimum biochar yield. Renewable and Sustainable Energy Reviews. 2022;167:112715.
- Zhou Y, Qin S, Verma S, Sar T, Sarsaiya S, Ravindran B, et al. Production and beneficial impact of biochar for environmental application: A comprehensive review. Bioresource Technology. 2021;337:125451.
- Sun J, Norouzi O, Mašek O. A state-of-the-art review on algae pyrolysis for bioenergy and biochar production. Bioresource technology. 2021:126258.
- Shirvanimoghaddam K, Czech B, Abdikheibari S, Brodie G, Kończak M, Krzyszczak A, et al. Microwave synthesis of biochar for environmental applications. Journal of Analytical and Applied Pyrolysis. 2021:105415.
- Patra BR, Mukherjee A, Nanda S, Dalai AK. Biochar production, activation and adsorptive applications: a review. Environmental Chemistry Letters. 2021;19(3):2237-59.
- Ghodake GS, Shinde SK, Kadam AA, Saratale RG, Saratale GD, Kumar M, et al. Review on biomass feedstocks, pyrolysis mechanism and physicochemical properties of biochar: State-of-the-art framework to speed up vision of circular bioeconomy. Journal of Cleaner Production. 2021;297:126645.
- Chun Y, Lee SK, Yoo HY, Kim SW. Recent Advancements in Biochar Production According to Feedstock Classification, Pyrolysis Conditions, and Applications: A Review. BioResources. 2021;16(3).

Reviewer 2 Report
In the review article, "Biochar as a soil amendment for reducing greenhouse gas emissions and improving soil carbon sink in arid and semi-arid regions: Current situation and ways forward," the authors attempt to summarize the impact of biochar on GHG emissions as well as the most effective factors related to the characteristics of biochar. Overall, it's well-structured and has been written properly. But there is still two points!
1. The title limits the world view of your work! while you're reviewing publications across the world. Indeed, there is an inconsistency between title and whole text. There is nothing in text that specialized your work just for arid and semi-arid regions. So, either change the title according to the text or the text according to the title. I think the first way is more logical!
2. There are too many references which are out of date! For example, more than 20 references older than 2010! While there are too many new publications that we can't neglect them. So Please replace old references with new ones. Also add these new published references which are matched with your scope:
Line 129-150: Regard to human activity, here is a new published work about global potential of abandoned sewage sludge in GHG emission and biochar processing. Also add this:
· 10.3390/ijerph191912983
line 562: Add a new published meta-analysis about changes in surface characteristics of biochar due activation:
· 10.1002/ldr.4464
Author Response
Responses to Reviewers' comments
We appreciate the valuable comments from the reviewers, which improved the revised version of our manuscript. We provided a full consideration for these comments, and detailed amendments are listed below in our point-by-point response. Reviewer's comments are highlighted in a blue color, and the response to these comments is presented in a black color. In addition, changes are highlighted in a red-color style in the revised version of the manuscript.
Authors' responses to Reviewer 2's comments:
Abstract
Comment 1: In the review article, "Biochar as a soil amendment for reducing greenhouse gas emissions and improving soil carbon sink in arid and semi-arid regions: Current situation and ways forward," the authors attempt to summarize the impact of biochar on GHG emissions as well as the most effective factors related to the characteristics of biochar. Overall, it's well-structured and has been written properly. But there is still two points!
Response 1: Thank you for the laud; we have revised and updated the manuscript as mentioned below point by point.
Comment 2: The title limits the world view of your work! while you're reviewing publications across the world. Indeed, there is an inconsistency between title and whole text. There is nothing in text that specialized your work just for arid and semi-arid regions. So, either change the title according to the text or the text according to the title. I think the first way is more logical!
Response 2: We appreciate this important comment. We changed the title of the manuscript by deleting “arid and semi-arid regions” to make a consistency between title and whole text and to improve the world view of our work.
Comment 4: There are too many references which are out of date! For example, more than 20 references older than 2010!
Response 4: Many thanks for this comment. We updated most of these references in the revised version of manuscript. Some works published before 2010 were kept because they serve as important guides for the research topic.
Comment 5: While there are too many new publications that we can't neglect them. So, please replace old references with new ones. Also add these new published references which are matched with your scope:
Line 129-150: Regard to human activity, here is a new published work about global potential of abandoned sewage sludge in GHG emission and biochar processing. Also add this:
- 10.3390/ijerph191912983
line 562: Add a new published meta-analysis about changes in surface characteristics of biochar due activation:
- 10.1002/ldr.4464
Response 5: Thanks for the valuable recommended articles. We have included these references and other relevant and updated works in different parts of the manuscript (e.g. Line 496 and Lines 539-543).

Reviewer 3 Report
Dear Authors, I am honored to review your valued review manuscript. The content you chose is a significant topic and required to be investigated. In your study, I noticed some delineations from the target of your title and study. One of them is in Section 2, where you explain the impacts of GHG's on the sustainability of agriculture which is not your concentrated sub-topic. Another one was, some the data you collected belong to non arid or semiarid sites which are reducing the strength of your study. One more point I noticed was your subtitle related to methane release. The methane formation generally occurs under unoxic conditions which is created by over saturation of soil pores with water. That high amount of water saturation can not be experienced in arid or semiarid conditions and as it can clearly be seen in the text, in the current subtitle, all the study samples were given from paddy soils and rice soils which are from humid regions. Thus, from all the study, if can not be given sufficient study samples from arid or semi-arid fields, the CH4 release factor should be eliminated.
Additionally, please kindly check my reviewed copy of your ms.
Good luck.

Author Response
Responses to Reviewers' comments
We appreciate the valuable comments from the reviewers, which improved the revised version of our manuscript. We provided full consideration for these comments, and detailed amendments are listed below in our point-by-point response. In addition, changes are highlighted in a red-color style in the revised version of the manuscript.
Comment 1: Dear Authors, I am honored to review your valued review manuscript. The content you chose is a significant topic and required to be investigated.
Response 1: We appreciate your kind comments and valuable feedback about our manuscript.
Comment 2: In your study, I noticed some delineations from the target of your title and study. One of them is in Section 2, where you explain the impacts of GHG's on the sustainability of agriculture which is not your concentrated sub-topic.
Response 2: We appreciate this valuable feedback from the reviewer. This section was omitted from the modified version of manuscript.
Comment 3: Another one was, some the data you collected belong to non-arid or semiarid sites which are reducing the strength of your study.
Response 3: We changed the title of the manuscript by deleting “arid and semi-arid regions” to make a consistency between title and whole text and to improve the world view of our work.
Comment 4: One more point, I noticed was your subtitle related to methane release. The methane formation generally occurs under anoxic conditions which is created by over saturation of soil pores with water. That high amount of water saturation cannot be experienced in arid or semiarid conditions and as it can clearly be seen in the text, in the current subtitle, all the study samples were given from paddy soils and rice soils which are from humid regions. Thus, from all the study, if cannot be given sufficient study samples from arid or semi-arid fields, the CH4 release factor should be eliminated.
Response 4: Many thanks for this suggestion. As mentioned in the previous comment, we changed the title of the manuscript by deleting “arid and semi-arid regions” to make a consistency between the title and whole text of the manuscript.
Comment 5: Additionally, please kindly check my reviewed copy of your ms.
Response 4: The attached copy of our manuscript has been checked and all suggested corrections/amendments have been carried out in the revised version of manuscript.

Reviewer 4 Report
Comments
The structure of the paper which is entitled “Biochar as a soil amendment…” looks very strange. I think the paper is not suitable to be published in Sustainability.
Detailed comments are shown as follows:
² What is the novelty of this work? Your own analysis on a systematic review is very important for a scientific paper.
² What results can you get from the paper? In the abstract of the paper, only one sentence, as shown in line 23-25, seems like a “result”.
² The manuscript looks like a review. Nevertheless, no enough comment exists in the manuscript. So many references, amounting to 148, have been cited in the paper. Are the data cited from the references reliable or not? I think an in-depth discussion is anticipated to avoid misleading readers with the mistakes from the references.
² An essential part, Methods and Materials, is missed in the manuscript. What’s the meaning of the method “meta–analysis” (line 118) ?.
² Still many language errors exist in the paper. In line 685 of reference 7, the words “et al.” may be replaced with the last author of the paper, “Djekic I.”; in line 245, the words “Fig. 3: ” may be replaced with the words “Figure 3.”.
Author Response
Responses to Reviewers' comments
We appreciate the valuable comments from the reviewers, which improved the revised version of our manuscript. We provided full consideration for these comments, and detailed amendments are listed below in our point-by-point response. In addition, changes are highlighted in a red-color style in the revised version of the manuscript.
Comment 1: What is the novelty of this work? Your own analysis on a systematic review is very important for a scientific paper.
Response 1:
Thanks for making such an insightful observation. In this overview, we look at the current situation of biochar use and where we might be going with biochar's potential to mitigate greenhouse gas emissions on a wide scale. Literature was combed for information on biochar's ability to reduce emissions of carbon monoxide, nitrous oxide, and methane. Unfortunately, the literature showed conflicting results, as biochar has been shown to be highly functional in limiting GHGs emissions in short-term laboratory studies but to have either no effect or even negative effects in large-scale field experiments. It is our opinion that this type of critical analysis was under-researched in the past. As such, this review article discusses the limitations of biochar in reducing greenhouse gas emissions at a large scale and offers recommendations for improving its performance. This overview also emphasizes the critical importance of functionalizing contemporary biochars with a greater inhibitory effect to act as a hedge against GHGs emissions. We also made a number of suggestions for future research that could be used to develop a strategy for using biochar on a large scale in the field to reduce GHG emissions.
Comment 2:What results can you get from the paper? In the abstract of the paper, only one sentence, as shown in line 23-25, seems like a “result”.
Response 2: We appreciate this valuable feedback from the reviewer. This critical review analyzed results of research undertaken during the last decade regarding the potentiality of biochar application in reducing GHGs emissions from soil and organic manures. We provided the concluded result of these data in the abstract (lines 21-25).
Comment 3: The manuscript looks like a review. Nevertheless, no enough comment exists in the manuscript. So many references, amounting to 148, have been cited in the paper. Are the data cited from the references reliable or not? I think an in-depth discussion is anticipated to avoid misleading readers with the mistakes from the references.
Response 3: Thanks for the comment. Absolutely, this is a review article. This review searched several published articles including books, book chapters, research articles, review articles and proceedings. We have clarified our methodology to select relevant works from the literature between 2010 and 2022. These works were derived from global databases (e.g. Scopus, Web of Science, ProQuest, EBSCO and JSTOR). This review highlights the prominent findings of these articles rather than re-peating their final conclusions. To meet the reviewer’s comment, further discussions have been incorporated in the revised version of manuscript (lines 318-332 and lines 381-387)
Comment 4: An essential part, Methods and Materials, is missed in the manuscript. What’s the meaning of the method “meta–analysis” (line 118) ?
Response 3: The section “Bibliographic data collection” has been incorporated in the revised version of manuscript to illustrate the protocol of data research, extraction and analysis in certified literature (lines 105-122). Meta-analysis is a type of research involved using statistical methods for analyzing results extracted from several scientific studies addressing the same question to derive a pooled estimate closest to the unknown common truth based on how this error is perceived. Further details about this concept are outlined in:
Herrera Ortiz AF., Cadavid Camacho E, Cubillos Rojas J, Cadavid Camacho T, Zoe Guevara S, Tatiana Rincón Cuenca N, Vásquez Perdomo A, Del Castillo Herazo V, & Giraldo Malo R. A Practical Guide to Perform a Systematic Literature Review and Meta-analysis. Principles and Practice of Clinical Research. 2022;7(4):47–57. https://doi.org/10.21801/ppcrj.2021.74.6
Books, book chapters, research articles, review articles, and proceedings were all scoured for this review. All selected sources were written in English and published within the last decade (2010 onwards). All of these articles came straight from reputable sources (e.g. Scopus, Web of Science, ProQuest, EBSCO and JSTOR). The International Biochar Initiative (IBI), the United States Department of Agriculture (USDA), and the European Environment Agency all contributed to the credibility of our review by providing official reports, statistics, and proceedings. Use of Get-Data Graph Digitizer (ver. 2.22, Russian Federation) allowed us to convert the data visualizations into corresponding numerical values. In this review, biochar + carbon dioxide emissions, biochar + nitrogen oxide emissions, biochar + chlorofluorocarbon emissions, biochar + soil + carbon dioxide emissions, biochar + soil + nitrogen oxide emissions, and biochar + soil + chlorofluorocarbon emissions were the initial search keywords. In addition, a number of meta-analyses were reported from a variety of published articles to establish a solid assessment of the current state of the potential use of biochar for limiting GHGs emissions and the prospects for this direction.
Comment 4: Still many language errors exist in the paper. In line 685 of reference 7, the words “et al.” may be replaced with the last author of the paper, “Djekic I.”; in line 245, the words “Fig. 3: ” may be replaced with the words “Figure 3.”.
Response 4: Many thanks for this suggestion. Typos and English errors were corrected in the revised version of manuscript. The last name “Djekic I.” has been involved in the reference in the updated version of manuscript.

Round 2
Reviewer 3 Report
Dear Authors, I am very pleased to see your corrections. I still have some concerns about your manuscript. Conversion of your focus from arid and semiarid lands to global ecosystems will make your paper much more comprehensive and cited following the publication. But remaining in your previous figure 2, Table 1 gives data for only arid and semiarid lands reveals missing data that you are attempting to give a global effect of biochar.
Additionally, Your focus on biochar production effects on GHGs seems that you handle only one side of the production of biochar effects on GHGs. You state in your section 4 the effects of feedstock and pyrolysis temperature effects on the capacity of biochar on reducing the GHGs. As you state here and you provide close data on the production effects of biochar is missing in your manuscript. I kindly suggest you interpret your data to reveal a better-comprehended paper, please put some data on the GHG released during the production of biochar.
In section 4.1. Please give a more precise explanation for biochar efficiency. Do you mean the application of biochar reduced to that ratio of the GHGs compared to non-biochar applications? Or are you intending to give example for the feedstocks when they are consumed in regular ignition they release GHGs at that higher amount of ratios?
And you can kindly combine the section 4.3. with previously provided section 3.1 titled 3.1. Effect of biochar application on reducing CO2 emissions whereas your title 4.3 was 4.3. Effect of application rate.
With my kindest regards.

Author Response
Responses to Reviewers' comments
We appreciate the valuable comments from the reviewers, which improved the second revised version of our manuscript. We provided full consideration for these comments, and detailed amendments are listed below in our point-by-point response. In addition, changes are highlighted in a red-color style in the second revised version of the manuscript.
Comment 1: Dear Authors, I am very pleased to see your corrections.
Response 1: We appreciate your valuable comments in both evaluation rounds.
Comment 2: I still have some concerns about your manuscript. Conversion of your focus from arid and semiarid lands to global ecosystems will make your paper much more comprehensive and cited following the publication. But remaining in your previous figure 2, Table 1 gives data for only arid and semiarid lands reveals missing data that you are attempting to give a global effect of biochar.
Response 2: Many thanks for this important suggestion. Both of Table 1 and Fig. 2 were updated in the revised version of manuscript by incorporating data of field investigations from different climatic zones.
Comment 3: Additionally, your focus on biochar production effects on GHGs seems that you handle only one side of the production of biochar effects on GHGs. You state in your section 4 the effects of feedstock and pyrolysis temperature effects on the capacity of biochar on reducing the GHGs. As you state here and you provide close data on the production effects of biochar is missing in your manuscript. I kindly suggest you interpret your data to reveal a better-comprehended paper, please put some data on the GHG released during the production of biochar. .
Response 3: Firstly, we appreciate this important suggestion. Indeed, studying GHGs emission during bichar production could be an interesting research point with high novelty aspects. This research point should be addressed as a main research point in the future research. However, we focus in this manuscript on the potentiality of biochar to offset GHGs from soil upon organic matter decomposition. We rephrased this sentence in the abstract to avoid any potential ambiguous in meaning (page 1, lines 14-16).
Comment 4: In section 4.1. Please give a more precise explanation for biochar efficiency. Do you mean the application of biochar reduced to that ratio of the GHGs compared to non-biochar applications? Or are you intending to give example for the feedstocks when they are consumed in regular ignition they release GHGs at that higher amount of ratios?
Response 4: Many thanks for this important point. Absolutely, this topic evaluates efficiency of biochar application on restraining GHGs relative to the unamended soil. As mentioned earlier, this review focuses on the efficiency of biochar in restraining GHGs emission from soil. The carbon footprint of feedstock during the thermochemical conversion of biochar is not our research topic in this manuscript. We rephrased this sentence in the revised version of manuscript to be more understandable for readers (page 12, lines 314-315). In section 4.1., we discussed factors related to feedstock type, which might affect the efficacy of biochar in reducing GHGs emissions (e.g. C/N ratio, aromaticity and stability against microbial decomposition)
Comment 5: And you can kindly combine the section 4.3. with previously provided section 3.1 titled 3.1. Effect of biochar application on reducing CO2 emissions whereas your title 4.3 was 4.3. Effect of application rate. .
Response 4: Thanks for this suggestion. However, the manuscript discusses in section "3.1." the concept of soil biochar application in restraining CO2 emissions and mechanisms involved in this regard. However, the manuscript discusses in section "4.3." different topic as it evaluates the effect of soil application rate of biochar in reducing GHGs emissions (CO2, N2O and CH4).
The attached copy of our manuscript has been checked and all suggested corrections/amendments have been carried out in the revised version of manuscript.

Reviewer 4 Report
The paper has been improved after the revision. Nevertheless, still I have not found enough novelties and comments.
Next time, a document including the revision notes is essential together with the revised paper.
Author Response
Responses to Reviewers' comments
We appreciate the valuable comments from the reviewers, which improved the second revised version of our manuscript. We provided full consideration for these comments, and detailed amendments are listed below in our point-by-point response. In addition, changes are highlighted in a red-color style in the second revised version of the manuscript.
Comment 1: The paper has been improved after the revision. Nevertheless, still I have not found enough novelties and comments.
Response 1:
Thanks for the positive feedback. Lots of research has introduced the high efficacy of biochar in restraining GHGs emissions under lab-scale experimentations. However, little has discussed the limitations of biochar in reducing GHGs emissions under field-scale investigations. The novelty of this review relies on assessing the key-factors that might impede the functionality of biochar toward restraining GHGs emissions. In addition, this review highlights the urgent need to develop fit-for-purpose forms of functionalized biochars to be tailored in improving soil carbon and restraining GHGs emissions from soil matrix. This review also evaluates the efficiency of biochar in reducing CO2 and N2O footprints of compost and other organic fertilizers during the microbial degradation stage. Finally, this review introduces a number of suggestions for the foreseeable research that could be used to develop a strategy for biochar utilization under field conditions to offset GHG emissions. The novelty of this review article has been introduced in the revised version of manuscript (page 2, lines 80-84).
Comment 2: Next time, a document including the revision notes is essential together with the revised paper.
Response 2: A document including the revision notes has been submitted alongside with the revised version of the manuscript.

Round 3
Reviewer 3 Report
Dear Authors,
I appreciate your collaboration to revise your valued paper and your extensive efforts to improve your manuscript. In its current status, your manuscript can be published as it is. With my kind regards.
Author Response
We appreciate the valuable efforts provided by the reviewer during the process of evaluation.